# Exploring patterns enriched in a dataset with contrastive principal component analysis

Abubakar Abid[1], Martin J. Zhang[1], Vivek K. Bagaria[1] & James Zou[2,3]

Visualization and exploration of high-dimensional data is a ubiquitous challenge across disciplines. Widely used techniques such as principal component analysis (PCA) aim to identify dominant trends in one dataset. However, in many settings we have datasets collected under different conditions, e.g., a treatment and a control experiment, and we are interested in visualizing and exploring patterns that are specific to one dataset. This paper proposes a method, contrastive principal component analysis (cPCA), which identifies low-dimensional structures that are enriched in a dataset relative to comparison data. In a wide variety of experiments, we demonstrate that cPCA with a background dataset enables us to visualize dataset-specific patterns missed by PCA and other standard methods. We further provide a geometric interpretation of cPCA and strong mathematical guarantees. An implementation of cPCA is publicly available, and can be used for exploratory data analysis in many applications where PCA is currently used.

[1] Department of Electrical Engineering, Stanford University, 450 Serra Mall, Stanford, CA 94305, USA. [2] Department of Biomedical Data Science, Stanford University, 450 Serra Mall, Stanford, CA 94305, USA. [3] Chan-Zuckerberg Biohub, 499 Illinois St., San Francisco, CA 94158, USA. These authors contributed equally: Abubakar Abid, Martin J. Zhang. Correspondence and requests for materials should be addressed to J.Z. (email: jamesz@stanford.edu)

Principal component analysis (PCA) is one of the most widely used methods for data exploration and visualization[1]. PCA projects the data onto a low-dimensional space and is especially powerful as an approach to visualize patterns, such as clusters, clines, and outliers in a dataset[2]. There is a large number of related visualization methods; for example, t-SNE[3] and multi-dimensional scaling (MDS)[4] allow for nonlinear data projections and may better capture nonlinear patterns than PCA. Yet, all of these methods are designed to explore one dataset at a time. When the analyst has multiple datasets (or multiple conditions in one dataset to compare), then the current state-of-practice is to perform PCA (or t-SNE, MDS, etc.) on each dataset separately, and then manually compare the various projections to explore if there are interesting similarities and differences across datasets[5,6]. Contrastive PCA (cPCA) is designed to fill in this gap in data exploration and visualization by automatically identifying the projections that exhibit the most interesting differences across datasets. Figure 1 provides an overview of cPCA that we explain in more detail ahead.

cPCA is motivated by a broad range of problems across disciplines. For illustration, we mention two such problems here and demonstrate others through experiments later in the paper. First, consider a dataset of gene-expression measurements from individuals of different ethnicities and sexes. This data includes gene-expression levels of cancer patients $\{\mathbf{x}_i\}$, which we are interested in analyzing. We also have control data, which corresponds to the gene-expression levels of healthy patients $\{\mathbf{y}_i\}$ from a similar demographic background. Our goal is to find trends and variations within cancer patients (e.g., to identify molecular subtypes of cancer). If we directly apply PCA to $\{\mathbf{x}_i\}$, however, the top principal components may correspond to the demographic variations of the individuals instead of the subtypes of cancers because the genetic variations due to the former are likely to be larger than that of the latter[7]. We approach this problem by noting that the healthy patients also contain the variation associated with demographic differences, but not the variation corresponding to subtypes of cancers. Thus, we can search for components in which $\{\mathbf{x}_i\}$ has high variance but $\{\mathbf{y}_i\}$ has low variance.

As a related example, consider a dataset $\{\mathbf{x}_i\}$ that consists of handwritten digits on a complex background, such as different images of grass (see Fig. 2(a), top). The goal of a typical unsupervised learning task may be to cluster the data, revealing the different digits in the image. However, if we apply standard PCA

on these images, we find that the top principal components do not represent features related to the handwritten digits, but reflect the dominant variation in features related to the image background (Fig. 2(b), top). We show that it is possible to correct for this by using a reference dataset $\{\mathbf{y}_i\}$ that consists solely of images of the grass (not necessarily the same images used in $\{\mathbf{x}_i\}$ but having similar covariance between features, as shown in Fig. 2(a), bottom), and looking for the subspace of higher variance in $\{\mathbf{x}_i\}$ compared to $\{\mathbf{y}_i\}$. By projecting onto this subspace, we can actually visually separate the images based on the value of the handwritten digit (Fig. 2(b), bottom). By comparing the principal components discovered by PCA with those discovered by cPCA, we see that cPCA identifies more relevant features (Fig. 2(c)), which allows us to use cPCA for such applications as feature selection and denoising[8].

Contrastive PCA is a tool for unsupervised learning, which efficiently reduces dimensionality to enable visualization and exploratory data analysis. This separates cPCA from a large class of supervised learning methods whose primary goal is to classify or discriminate between various datasets, such as linear discriminant analysis (LDA)[9], quadratic discriminant analysis (QDA)[10], supervised PCA[11], and QUADRO[12]. This also distinguishes cPCA from methods that integrate multiple datasets[13–16], with the goal of identifying correlated patterns among two or more datasets, rather than those unique to each individual dataset. There is also a rich family of unsupervised methods for dimension reduction besides PCA. For example, multi-dimensional scaling (MDS)[4] finds a low-dimensional embedding that preserves the distance in the high-dimensional space; principal component pursuit[17] finds a low-rank subspace that is robust to small entry-wise noise and gross sparse errors. But none are designed to utilize relevant information from a second dataset, as cPCA does. In the supplement, we have compared cPCA to many of the previously-mentioned techniques on representative datasets (see Supplementary Figs. 3 and 4).

In a specific application domain, there may be specialized tools in that domain with similar goals as cPCA[18–20]. For example, in the results, we show how cPCA applied on genotype data visualizes geographical ancestry within Mexico. Exploring fine-grained clusters of genetic ancestries is an important problem in population genetics, and researchers have recently developed an algorithm to specifically visualize such ancestry clusters[18]. While cPCA performs well here, the expert-crafted algorithm might perform even better for a specific dataset. However, the

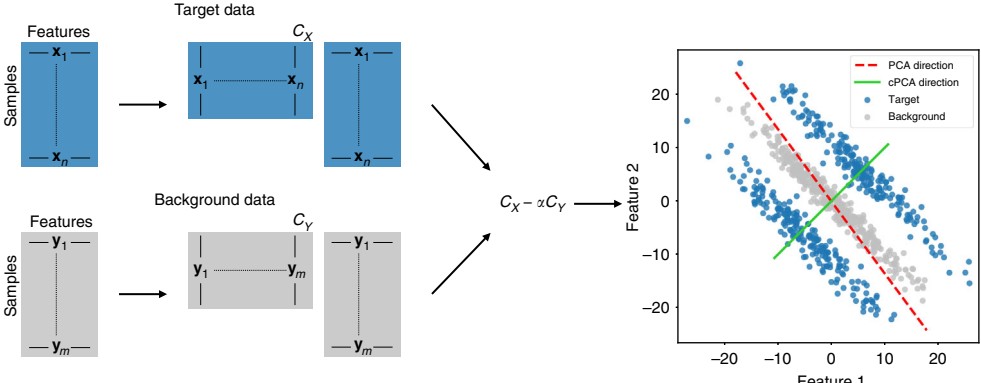

**Fig. 1** Schematic Overview of cPCA. To perform cPCA, compute the covariance matrices $C_X$, $C_Y$ of the target and background datasets. The singular vectors of the weighted difference of the covariance matrices, $C_X - \alpha \cdot C_Y$, are the directions returned by cPCA. As shown in the scatter plot on the right, PCA (on the target data) identifies the direction that has the highest variance in the target data, while cPCA identifies the direction that has a higher variance in the target data as compared to the background data. Projecting the target data onto the latter direction gives patterns unique to the target data and often reveals structure that is missed by PCA. Specifically, in this example, reducing the dimensionality of the target data by cPCA would reveal two distinct clusters

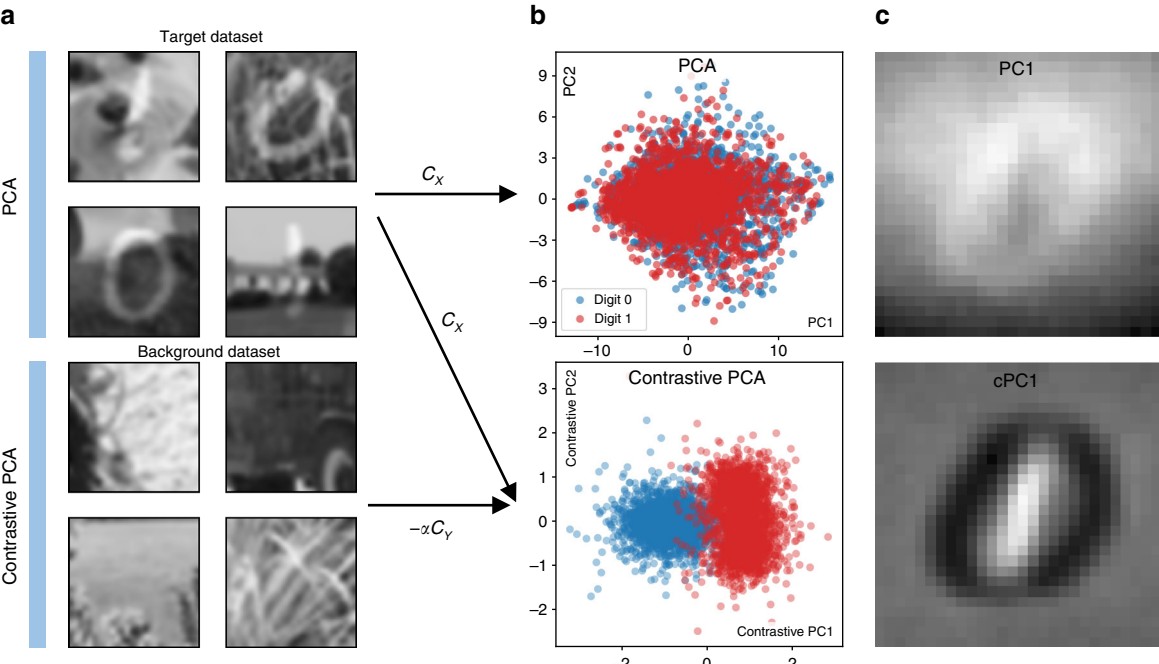

**Fig. 2** Contrastive PCA on Noisy Digits. **a**, Top: We create a target dataset of 5,000 synthetic images by randomly superimposing images of handwritten digits 0 and 1 from MNIST dataset[32] on top of images of grass taken from ImageNet dataset[33] belonging to the synset grass. The images of grass are converted to grayscale, resized to be 100 × 100, and then randomly cropped to be the same size as the MNIST digits, 28 × 28. **b**, Top: Here, we plot the result of embedding the synthetic images onto their first two principal components using standard PCA. We see that the points corresponding to the images with 0's and images with 1's are hard to distinguish. **a**, Bottom: A background dataset is then introduced consisting solely of images of grass belonging to the same synset, but we use images that are different than those used to create the target dataset. **b**, Bottom: Using cPCA on the target and background datasets (with a value of the contrast parameter $\alpha$ set to 2.0), two clusters emerge in the lower-dimensional representation of the target dataset, one consisting of images with the digit 0 and the other of images with the digit 1. **c** We look at the relative contribution of each pixel to the first principal component (PC) and first contrastive principal component (cPC). Whiter pixels are those that carry a more positive weight, while darker denotes those pixels that carry negative weights. PCA tends to emphasize pixels in the periphery of the image and slightly de-emphasize pixels in the center and bottom of the image, indicating that most of the variance is due to background features. On the other hand, cPCA tends to upweight the pixels that are at the location of the handwritten 1's, negatively weight pixels at the location of handwritten 0's, and neglect most other pixels, effectively discovering those features useful for discriminating between the superimposed digits

specialized algorithm requires substantial domain knowledge to design, is more computationally expensive, and can be challenging to use. The goal of cPCA is not to replace all these specialized state-of-the-art methods in each of their domains, but to provide a general method for exploring arbitrary datasets.

We propose a concrete and efficient algorithm for cPCA in this paper. The method takes as input a target dataset $\{\mathbf{x}_i\}$ that we are interested in visualizing or identifying patterns within. As a secondary input, cPCA takes a background dataset $\{\mathbf{y}_i\}$, which does not contain the patterns of interest. The cPCA algorithm returns subspaces which capture a large amount of variation in the target data $\{\mathbf{x}_i\}$, but little in the background $\{\mathbf{y}_i\}$ (see Fig. 1, Methods, and Supplementary Methods for more details). This subspace corresponds to features containing structure specific to $\{\mathbf{x}_i\}$. Hence, when the target data is projected onto this subspace, we are able to visualize and discover the additional structure in the target data relative to the background. Analogous to the principal components (PCs), we call the directions found by cPCA the contrastive principal components (cPCs). We emphasize that cPCA is fundamentally an unsupervised technique, designed to resolve patterns in one dataset more clearly by using the background dataset as a contrast. In particular, cPCA does not seek to discriminate between the target and background datasets; the subspace that contains trends that are enriched in the target dataset is not necessarily the same subspace that is optimal for classification between the datasets.

## Results

**Subgroup discovery in protein expression data**. Researchers have noted that standard PCA is often ineffective at discovering subgroups within biological data, at least in part because "dominant principal components…correlate with artifacts,"[21] rather than with features that are of interest to the researcher. How can cPCA be used in these settings to detect the more significant subgroups? By using a background dataset to cancel out the universal but uninteresting variation in the target, we can search for structure that is unique to the target dataset.

Our first experiment uses a dataset consisting of protein expression measurements of mice that have received shock therapy[22,23]. Some of the mice have developed Down Syndrome (DS). To create an unsupervised learning task where we have ground truth information to evaluate the methods, we assume this DS information is not known to the analyst and only use it for algorithm evaluation. We would like to see if we detect any significant differences within the shocked mice population in an unsupervised manner (the presence or absence of Down Syndrome being a key example). In Fig. 3a (top), we show the result of applying PCA to the target dataset: the transformed data does not reveal any significant clustering within the population of mice. The major sources of variation within mice may be natural, such as sex or age.

We apply cPCA to this dataset using a background consists of protein expression measurements from a set of mice that have not

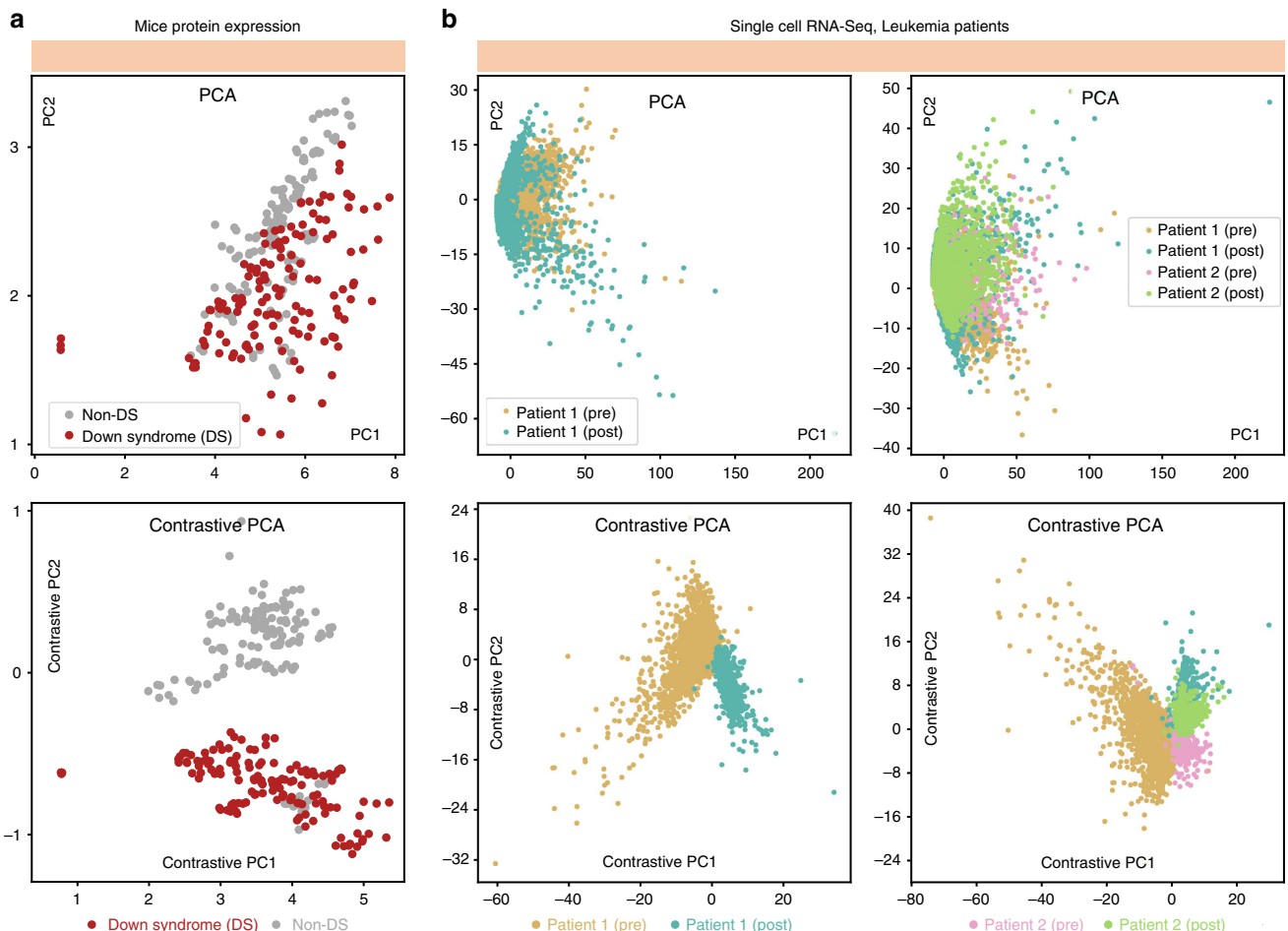

**Fig. 3** Discovering subgroups in biological data. **a** We use PCA to project a protein expression dataset of mice with and without Down Syndrome (DS) onto the first two components. The lower-dimensional representation of protein expression measurements from mice with and without DS are seen to be distributed similarly (top). But, when we use cPCA to project the dataset onto its first two cPCs, we discover a lower-dimensional representation that clusters mice with and without DS separately (bottom). **b** Furthermore, we use PCA and cPCA to visualize a high-dimensional single-cell RNA-Seq dataset in two dimensions. The dataset consists of four cell samples from two leukemia patients: a pre-transplant sample from patient 1, a post-transplant sample from patient 1, a pre-transplant sample from patient 2, and a post-transplant sample from patient 2. **b**, left: The results using only the samples from patient 1, which demonstrate that cPCA (bottom) more effectively separates the samples than PCA (top). When the samples from the second patient are included, in **b**, right, again cPCA (bottom) is more effective than PCA (top) at separating the samples, although the post-transplant cells from both patients are similarly-distributed. We show plots of each sample separately in Supplementary Fig. 5, where it is easier to see the overlap between different samples

been exposed to shock therapy. They are control mice that likely have similar natural variation as the experimental mice, but without the differences that result from the shock therapy. With this dataset as a background, cPCA is able to resolve two different groups in the transformed target data, one corresponding to mice that do not have Down Syndrome and one corresponding (mostly) to mice that have Down Syndrome, as illustrated in Fig. 3a (bottom). As a comparison, we also applied 8 other dimensionality reduction techniques to identify directions that differentiate between the target and background datasets, none of which were able to separate the mice as well as cPCA (see Supplementary Fig. 4 for details).

**Subgroup discovery in single-cell RNA-Seq data**. Next, we analyze a higher-dimensional public dataset consisting of single-cell RNA expression levels of a mixture of bone marrow mono-nuclear cells (BMMCs) taken from a leukemia patient before stem-cell transplant and BMMCs from the same patient after stem-cell transplant[24]. All single-cell RNA-Seq data is pre-processed using similar methods as described by the authors. In

particular, before applying PCA or cPCA, all datasets are reduced to 500 genes, which are selected on the basis of highest dispersion [variance divided by mean] within the target data. Again, we perform PCA to see if we can visually discover the two samples in the transformed data. As shown in Fig. 3b (top left), both cell types follow a similar distribution in the space spanned by the first two PCs. This is likely because the differences between the samples is small and the PCs instead reflect the heterogeneity of various kinds of cells within each sample or even variations in experimental conditions, which can have a significant effect on single-cell RNA-Seq measurements[25].

We apply cPCA using a background dataset that consists of RNA-Seq measurements from a healthy individual's BMMC cells. We expect that this background dataset to contain the variation due to the heterogeneous population of cells as well as variations in experimental conditions. We may hope, then, that cPCA might be able to recover directions that are enriched in the target data, corresponding to pre- and post-transplant differences. Indeed, that is what we find, as shown in Fig. 3b (bottom left).

We further augment our target dataset with BMMC samples from a second leukemia patient, again before and after stem-cell

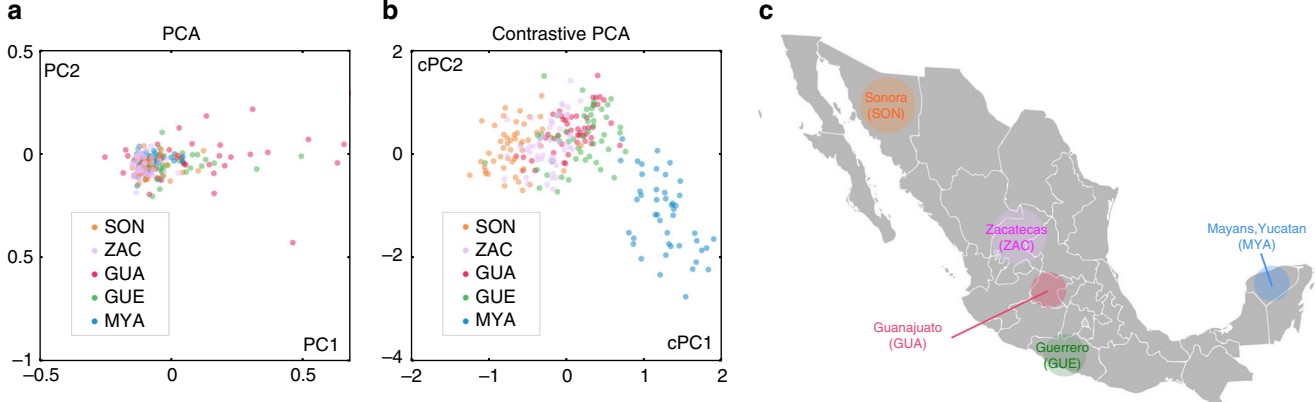

**Fig. 4** Relationship between Mexican ancestry groups. **a** PCA applied to genetic data from individuals from 5 Mexican states does not reveal any visually discernible patterns in the embedded data. **b** cPCA applied to the same dataset reveals patterns in the data: individuals from the same state are clustered closer together in the cPCA embedding. **c** Furthermore, the distribution of the points reveals relationships between the groups that matches the geographic location of the different states: for example, individuals from geographically adjacent states are adjacent in the embedding. **c** Adapted from a map of Mexico that is originally the work of User:Allstrak at Wikipedia, published under a CC-BY-SA license, sourced from https://commons.wikimedia.org/wiki/File:Mexico_Map.svg

transplant. Thus, there are a total of four subpopulations of cells. Application of PCA on this data shows that the four subpopulations are not separable in the subspace spanned by the top two principal components (PCs), as shown in Fig. 3b (top right). Again, however, when cPCA is applied with the same background dataset, at least three of the subpopulations show much stronger separation, as shown in Fig. 3b (bottom right). The cPCA embedding also suggests that the cell samples from both patients are more similar to each other after stem-cell transplant (cyan and green dots) than before the transplant (gold and pink dots), a reasonable hypothesis which can be tested by the investigator. One may refer to Supplementary Fig. 5 for more details of this experiment. We see that cPCA can be a useful tool to infer the relationship between subpopulations, a topic we explore further next.

**Relationship between ancestral groups in Mexico.** In previous examples, we have seen that cPCA allows the user to discover subclasses within a target dataset that are not labeled a priori. However, even when subclasses are known ahead of time, dimensionality reduction can be a useful way to visualize the relationship within groups. For example, PCA is often used to visualize the relationship between ethnic populations based on genetic variants, because projecting the genetic variants onto two dimensions often produces maps that offer striking visualizations of geographic and historic trends[26,27]. But again, PCA is limited to identifying the most dominant structure; when this represents universal or uninteresting variation, cPCA can be more effective at visualizing trends.

The dataset that we use for this example consists of single nucleotide polymorphisms (SNPs) from the genomes of individuals from five states in Mexico, collected in a previous study[28]. Mexican ancestry is challenging to analyze using PCA since the PCs usually do not reflect geographic origin within Mexico; instead, they reflect the proportion of European/Native American heritage of each Mexican individual, which dominates and obscures differences due to geographic origin within Mexico (see Fig. 4a). To overcome this problem, population geneticists manually prune SNPs, removing those known to derive from Europeans ancestry, before applying PCA. However, this procedure is of limited applicability since it requires knowing the origin of the SNPs and that the source of background

variation to be very different from the variation of interest, which are often not the case.

As an alternative, we use cPCA with a background dataset that consists of individuals from Mexico and from Europe. This background is dominated by Native American/European variation, allowing us to isolate the intra-Mexican variation in the target dataset. The results of applying cPCA are shown in Fig. 4b. We find that individuals from the same state in Mexico are embedded closer together. Furthermore, the two groups that are the most divergent are the Sonorans and the Mayans from Yucatan, which are also the most geographically distant within Mexico, while Mexicans from the other three states are close to each other, both geographically as well as in the embedding captured by cPCA (see Fig. 4c). See also Supplementary Fig. 6 for more details.

## Discussion
In many data science settings, we are interested in visualizing and exploring patterns that are enriched in one dataset relative to other data. We have presented cPCA as a general tool for performing such contrastive exploration, and we have illustrated its usefulness in a diverse range of applications. The main advantages of cPCA are its generality and ease of use. Computing a particular cPCA takes essentially the same amount of time as computing a regular PCA. This computational efficiency enables cPCA to be useful for interactive data exploration, where each operation should ideally be almost immediate. As such, any settings where PCA is applied on related datasets, cPCA can also be applied. In the Supplementary Note 3 and Supplementary Fig. 8, we show how cPCA can be kernelized to uncover nonlinear contrastive patterns in datasets.

The only free parameter of contrastive PCA is the contrast strength $\alpha$. In our default algorithm, we developed an automatic scheme based on clusterings of subspaces for selecting the most informative values of $\alpha$ (see Methods). All of the experiments performed for this paper use the automatically generated $\alpha$ values, and we believe this default will be sufficient in many applications of cPCA. The user may also input specific values for $\alpha$ if more fine-grained exploration is desired.

cPCA, like regular PCA and other dimensionality reduction methods, does not give $p$-values or other statistical significance quantifications. The patterns discovered through cPCA need to be validated through hypothesis testing or additional analysis using

relevant domain knowledge. We have released the code for cPCA as a python package along with documentation and examples.

## Methods

**Description of the Algorithm**. For the $d$-dimensional target data $\{\mathbf{x}_i \in \mathbb{R}^d\}$ and background data $\{\mathbf{y}_i \in \mathbb{R}^d\}$, let $C_X$, $C_Y$ be their corresponding empirical covariance matrices. Let $\mathbb{R}^d_{\mathrm{unit}} \stackrel{\text{def}}{=} \{\mathbf{v} \in \mathbb{R}^d : \|\mathbf{v}\|_2 = 1\}$ be the set of unit vectors. For any direction $\mathbf{v} \in \mathbb{R}^d_{\mathrm{unit}}$, the variance it accounts for in the target data and in the background data can be written as:

$$\text{Target data variance}: \quad \lambda_X(\mathbf{v}) \stackrel{\text{def}}{=} \mathbf{v}^T C_X \mathbf{v},$$

$$\text{Background data variance}: \quad \lambda_Y(\mathbf{v}) \stackrel{\text{def}}{=} \mathbf{v}^T C_Y \mathbf{v}.$$

Given a contrast parameter $\alpha \geq 0$ that quantifies the trade-off between having high target variance and low background variance, cPCA computes the contrastive direction $\mathbf{v}^*$ by optimizing

$$\mathbf{v}^* = \operatorname{argmax}_{\mathbf{v} \in \mathbb{R}^d_{\mathrm{unit}}} \lambda_X(\mathbf{v}) - \alpha\lambda_Y(\mathbf{v}). \tag{1}$$

This problem can be rewritten as

$$\mathbf{v}^* = \operatorname{argmax}_{\mathbf{v} \in \mathbb{R}^d_{\mathrm{unit}}} \mathbf{v}^T(C_X - \alpha C_Y)\mathbf{v},$$

which implies that $\mathbf{v}^*$ corresponds to the first eigenvector of the matrix $C \stackrel{\text{def}}{=} (C_X - \alpha C_Y)$. Hence the contrastive directions can be efficiently computed using eigenvalue decomposition. Analogous to PCA, we call the leading eigenvectors of $C$ the contrastive principal components (cPCs). We note the cPCs are eigenvectors of the matrix $C$ and are hence orthogonal to each other. For a fixed $\alpha$, we compute (1) and return the subspace spanned by the first few (typically two) cPCs.

The contrast parameter $\alpha$ represents the trade-off between having the high target variance and the low background variance. When $\alpha = 0$, cPCA selects the directions that only maximize the target variance, and hence reduces to PCA applied on the target data $\{\mathbf{x}_i\}$. As $\alpha$ increases, directions with smaller background variance become more important and the cPCs are driven towards the null space of the background data $\{\mathbf{y}_i\}$. In the limiting case $\alpha = \infty$, any direction not in the null space of $\{\mathbf{y}_i\}$ receives an infinite penalty. In this case, cPCA corresponds to first projecting the target data onto the null space of the background data, and then performing PCA on the projected data.

Instead of choosing a single $\alpha$ and returning its subspace, cPCA computes the subspaces of a list of $\alpha$'s and returns a few subspaces that are far away from each other in terms of the principal angle[29]. Projecting the data onto each of these subspaces will reveal different trends within the target data, and by visually examining the scatterplots that are returned, the user can quickly discern the relevant subspace (and corresponding value of $\alpha$) for his or her analysis. See Supplementary Fig. 1 for a detailed example.

The complete algorithm of cPCA is described in Algorithm 2 (Supplementary Methods). We typically set the list of potential values of $\alpha$ to be 40 values logarithmically spaced between 0.1 and 1000 and this is used for all experiments in the paper. To select the representative subspaces, cPCA uses spectral clustering[30] to cluster the subspaces, where the affinity is defined as the product of the cosine of the principal angles between the subspaces. Then the medoids (representative) of each cluster are used as the values of $\alpha$ to generate the scatterplots seen by the user.

**Choosing the background dataset**. The choice of the background dataset has a large influence on the result of cPCA. In general, the background data should have the structure that we would like to remove from the target data. Such structure usually corresponds to directions in the target with high variance, but that are not of interest to the analyst.

We provide a few general examples of background datasets that may provide useful contrasts to target data: (1) A control group $\{\mathbf{y}_i\}$ contrasted with a diseased population $\{\mathbf{x}_i\}$ because the control group contains similar population-level variation but not the subtle variation due to different subtypes of the disease. (2) The data at time zero $\{\mathbf{y}_i\}$ used to contrast against data at a later time point $\{\mathbf{x}_i\}$. This enables visualizations of the most salient changes over time. (3) A homogeneous group $\{\mathbf{y}_i\}$ contrasted with a mixed group $\{\mathbf{x}_i\}$ because both have intra-population variation and measurement noise, but the former does not have inter-population variation. (4) A pre-treatment dataset $\{\mathbf{y}_i\}$ contrasted with post-treatment data $\{\mathbf{x}_i\}$ to remove measurement noise but preserve variations caused by treatment. (5) A set of signal-free recordings $\{\mathbf{y}_i\}$ or images that contain only noise, contrasted with measurements $\{\mathbf{x}_i\}$ that consist of both signal and noise.

It is worth adding that the background data does not need to have exactly the same covariance structure as what we would like to remove from the target dataset. As an example, in the experiment shown in Fig. 2, it turns out that we do not need to use a background dataset that consists of images of grass. In fact, similar results are obtained even if instead of images of grass, images of the sky are used as the background dataset. As the structure of the covariance matrices are similar enough, cPCA removes the background structure from the target data. In addition, cPCA does not require the target data and the background data to have a

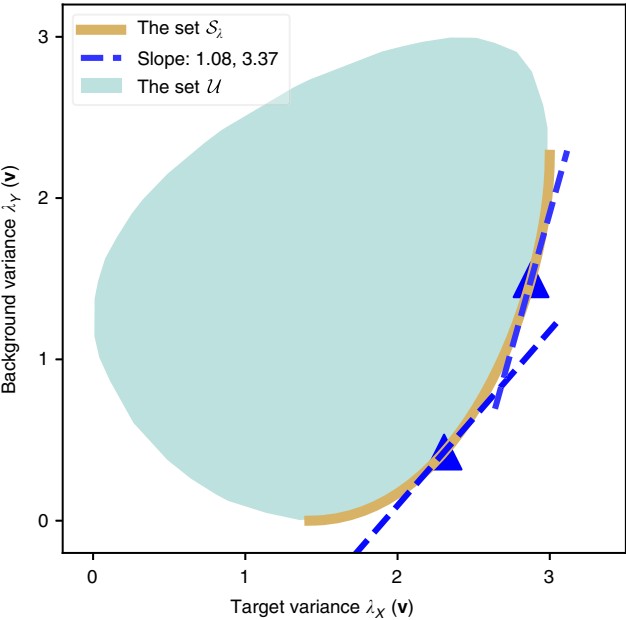

**Fig. 5** Geometric Interpretation of cPCA. The set of target–background variance pairs $\mathcal{U}$ is plotted as the teal region for some randomly generated target and background data. The lower-right boundary, as colored in gold, corresponds to the set of most contrastive directions $\mathcal{S}_\lambda$. The blue triangles are the variance pairs for the cPCs selected with $\alpha$ values 0.92 and 0.29 respectively. We note that they correspond to the points of tangency of the gold curve and the tangent lines with slope $\frac{1}{\alpha} = 1.08$, 3.37, respectively

similar number of samples. Since the covariance matrices are computed independently, cPCA only requires that the empirical covariance matrices be good estimates of the underlying population covariance matrices, essentially the same requirement as PCA.

**Theoretical guarantees of cPCA**. Here, we discuss the geometric interpretation of cPCA as well as its statistical properties. First, it is interesting to consider which directions are "better" for the purpose of contrastive analysis. For a direction $\mathbf{v} \in \mathbb{R}^d_{\mathrm{unit}}$, its significance in cPCA is fully determined by its target–background variance pair $(\lambda_X(\mathbf{v}), \lambda_Y(\mathbf{v}))$; it is desirable to have a higher target variance and a lower background variance. Based on this intuition, we can further define a partial order of contrastiveness for various directions: for two directions $\mathbf{v}_1$ and $\mathbf{v}_2$, we might say $\mathbf{v}_1$ is a better contrastive direction if it has a higher target variance and a lower background variance. In this case, the target–background variance pair of $\mathbf{v}_1$ would lie on the lower-right side of that of $\mathbf{v}_2$ in the plot of target–background variance pairs $(\lambda_X(\mathbf{v}), \lambda_Y(\mathbf{v}))$, e.g., Fig. 5. Based on this partial order, the set of most contrastive directions can be defined in a similar fashion as the definition of the Pareto frontier[31]. Let $\mathcal{U}$ be the set of target–background variance pairs for all directions, i.e. $\mathcal{U} \stackrel{\text{def}}{=} \{(\lambda_X(\mathbf{v}), \lambda_Y(\mathbf{v}))\}_{\mathbf{v} \in \mathbb{R}^d_{\mathrm{unit}}}$. The set of most contrastive directions corresponds to the lower-right boundary of $\mathcal{U}$ in the plot of target–background variance pairs, as shown in Fig. 5. (For the particular case of simultaneously diagonalizable background and target matrices, see Supplementary Fig. 7.)

Regarding cPCA, we can prove (see Supplementary Note 2) that by varying $\alpha$, the set of top cPC's is identical to the set of most contrastive directions. Moreover, for the direction $\mathbf{v}$ selected by cPCA with the contrast parameter set to $\alpha$, its variance pair $(\lambda_X(\mathbf{v}), \lambda_Y(\mathbf{v}))$ corresponds to the point of tangency of the lower-right boundary of $\mathcal{U}$ with a slope-$1/\alpha$ line. As a result, by varying $\alpha$ from zero to infinity, cPCA selects directions with variance pairs traveling from the lower-left end to the upper-right end of the lower-right boundary of $\mathcal{U}$.

We also remark that regarding the randomness of the data, the convergence rate of the sample cPC to the population cPC is $O_p\left(\sqrt{\frac{d}{\min(n,m)}}\right)$ under mild assumptions, where $d$ is the dimension and $n,m$ are the sizes of the target and the background data. This rate is similar to the standard convergence rate of the sample eigenvector for a covariance matrix. See Supplementary Note 2.

**Code availability**. We have released a Python implementation of contrastive PCA on GitHub (https://github.com/abidlabs/contrastive). The GitHub repository also includes Python notebooks and datasets that reproduce most of the figures in this paper and in the Supplementary Information.

**Data availability**. Datasets that have been used to evaluate contrastive PCA in this paper are either available from us or from the authors of the original studies. Please see the GitHub repository listed in the previous section for the datasets that we have released.

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

## Acknowledgements

We thank Alex Ioannidis for the assistance in carrying out the experiments on the relationship between acestral groups in Mexico. We thank Professor David Tse for providing helpful suggestions and financially supporting M.Z. and V.B. We thank our colleagues Amirata Ghorbani, Xinkun Nie, and Ruishan Liu for helpful comments in the development of this technique. A.A. and M.Z. are partially supported by Stanford Graduate Fellowship. J.Z. is supported by a Chan-Zuckerberg Investigator grant and by National Science Foundation grant CRII 1657155.

## Author contributions

J.Z. proposed the original notion of contrastive PCA and supervised the research. A.A., M.Z., and V.B. designed the algorithm. A.A. implemented the algorithm and carried out the empirical experiments. M.Z. and V.B. proved the theoretical results. A.A. and M.Z. wrote the manuscript. All of the authors reviewed the manuscript.

## Additional information

**Competing interests:** The authors declare no competing interests.

