## [Peer Review File · Nature Communications]

Reviewers' comments:

Reviewer #1 (Remarks to the Author):

In this article, the authors propose a method, entitled contrastive principal component analysis (cPCA), which performs a decomposition of the difference in two covariance matrices to identify features that are enriched in one dataset relative to a reference/control dataset. This is a supervised analysis that requires one to know the groups a priori. In the primary manuscript, it is compared to PCA. In the supplement, a brief comparison to LDA and FA is included (Fig S2).

The major claim of the paper is that cPCA outperforms PCA. However PCA (or FA) were not designed to find the components that discriminate between groups, therefore the finding that a method that is optimized to find the difference between 2 groups outperforms PCA or FA for that purpose is not surprising.

In the supplement there is a brief comparison of cPCA to LDA. There is no description of how LDA was implemented, if classic LDA was used it would be constrained by data dimensionality ($n \gg p$). Was the data filtered or what implementation of LDA was used. LDA presumes equal covariance matrices for both classes but different means. Quadratic discriminant analysis assumes different covariance and means across classes and might be a better comparator. A comparison to regularized QDA might be more appropriate. Equally there have been several supervised implementations of PCA, including from Tibshirani's group (available in several R packages including superPC), Barshan et al., 2011 (<https://doi.org/10.1016/j.patcog.2010.12.015>) or between group analysis (available in the ade4 R package, <https://www.ncbi.nlm.nih.gov/pubmed/12490444>). Please compare your approach to one or more supervised PCA approaches

This method performs a PCA of the difference in the covariance matrices. Does this approach work better than a PCA of results of a linear regression (eg limma). I am asking this as limma linear regression is widely performed by researchers to find features that separate two groups.

One of my principal concerns about this manuscript is the lack of reference to a rich resource of existing literature and methods in the field. Performing a decomposition of the differences between two matrices has been under-applied in genomics, however is widely used in other biological fields analysis. For example, distance based redundancy analysis, principal coordinates analysis and multiple dimensional scaling all are widely used (many implementations are described in classic textbook numerical ecological by Legendre & Legendre, 1998, or recently reviewed in Scientific Reports 7: 6481 (2017). doi:10.1038/s41598-017-06693-z). Please compare your approach distance based matrix decomposition approaches.

The idea of separating a background and foreground population in data is useful and has been widely applied. Arguably one of the most popular background-foreground matrix decomposition approaches is projection pursuit. Please compare your approach to projection pursuit.

There have been other dimension reduction approaches that employ the difference in the covariance matrix described in other fields. Please review if your method is similar to (QUADRO), Fan et al, Ann Stat. 2015; 43(4): 1498–1534.) and Jin et al., 2005 (<https://dl.acm.org/citation.cfm?id=2118631>).

How does homoscedasticity impact the performance of the approach? Is the approach robust to heteroscedasticity?

Reviewer #2 (Remarks to the Author):

This paper is well-written and very pleasant to read. I think it should be accepted after some minor revisions. My comments are following.

(1) Is there any probabilistic interpretation to the cPCA approach? Perhaps analogous to the probabilistic PCA model (see reference below).

Tipping, M. E., & Bishop, C. M. (1999). Probabilistic principal component analysis. *Journal of the Royal Statistical Society: Series B (Statistical Methodology)*, 61(3), 611-622.

(2) I appreciate that the authors made the implementation available. Regarding the reproducibility of the real data analysis, I wonder if the data sets and data analysis code can also be made available. Some of the analysis is difficult to reproduce exactly from the paper. For example, on page 7, "...before applying PCA or cPCA, all datasets are reduced to 500 genes, which are selected on the basis of highest dispersion within the target data".

(3) The authors claimed that a "strong mathematical guarantee for the method". But I do not find the theorems useful, since it failed to take into account any randomness in the data. In other words, I am wondering if there is any statistical guarantee like PCA or some other popular dimension reduction methods.

We thank the reviewers for their thoughtful feedback. We provide point-by-point response to all of the reviewers' questions below.

REVIEWER 1 COMMENTS:

1. In this article, the authors propose a method, entitled contrastive principal component analysis (cPCA), which performs a decomposition of the difference in two covariance matrices to identify features that are enriched in one dataset relative to a reference/control dataset. This is a supervised analysis that requires one to know the groups a priori. In the primary manuscript, it is compared to PCA. In the supplement, a brief comparison to LDA and FA is included (Fig S2).

We appreciate the reviewer's careful attention to comparisons of cPCA to other techniques. We would like to point out that cPCA is not a supervised technique, as cPCA is designed to explore patterns (like subgroups, outliers, and gradual trends) in a single *target* dataset via dimensionality reduction. In fact, cPCA uses a secondary *background* dataset only to facilitate its dimensionality reduction. We show that cPCA often finds a better low-dimensional representation of a target dataset than standard unsupervised techniques (like PCA) because it uses the fact that the background dataset does not contain the variation of interest.

We emphasize that the purpose of cPCA is *not* to classify data points between the two datasets, as the subspace that is optimal for classification is not necessarily the same subspace that contains trends that are enriched in the target dataset. To make this point clearer in the manuscript itself, we have amended the main text. We have added the following in the "Overview of the cPCA algorithm" section (pg. 4):

We emphasize that cPCA is fundamentally an unsupervised technique, designed to resolve patterns in one dataset more clearly by using the background dataset as a contrast. In particular, cPCA does *not* seek to discriminate between the target and background datasets; the subspace that contains trends that are enriched in the target dataset is not necessarily the same subspace that is optimal for classification between the datasets.

2. The major claim of the paper is that cPCA outperforms PCA. However PCA (or FA) were not designed to find the components that discriminate between groups, therefore the finding that a method that is optimized to find the difference between 2 groups outperforms PCA or FA for that purpose is not surprising.

The reviewer is correct that PCA and FA are not designed to discriminate between datasets. The purpose of cPCA is not to discriminate between datasets either. As we explained in our response above, the purpose of cPCA is to find patterns in a *single* dataset more effectively by using a background dataset as a contrast. The reason that we included comparisons to PCA and FA is because they demonstrate that cPCA, by using a background dataset, discovers patterns that standard dimensionality reduction techniques, using only the target dataset, do not discover. Please see the response to (4) below for more comprehensive comparisons to other techniques.

3. In the supplement there is a brief comparison of cPCA to LDA. There is no description of how LDA was implemented, if classic LDA was used it would be constrained by data dimensionality ($n \gg p$). Was the data filtered or what implementation of LDA was used.

We appreciate the reviewer’s concern for transparency. In the experiment, we used the *LinearDiscriminantAnalysis* class from *sklearn.discriminant_analysis* for comparison. We have released a python notebook that contains the data and code to replicate our experiments comparing contrastive PCA to other techniques, including LDA. This notebook can be accessed at the GitHub repository for this paper: <https://github.com/abidlabs/contrastive/tree/master/experiments>. With regards to the specific questions about LDA, as the reviewer points out, that the number of data points (n) needs to be larger than the dimensionality (p) to perform LDA. It is preferable that the number of data points is many times the dimensionality so that the empirical covariance matrix serves as a good estimation of the population covariance matrix. In our example of mice protein data, $n = 405$ and $p = 77$, so $n \approx 5p$.

Furthermore, to verify empirically that the covariance matrix had converged, we “bootstrapped” our dataset by drawing samples randomly with replacement from the original dataset to create 3 new datasets, each of the same size as the original. We ran PCA, cPCA, and LDA on these bootstrapped datasets, and found that the projected data were visually consistent across the datasets. We have included Fig. 1 here to show these results.

Figure 1: 2-dimensional representations of three bootstrapped versions of the mice protein dataset.

4. LDA presumes equal covariance matrices for both classes but different means. Quadratic discriminant analysis assumes different covariance and means across classes and might

be a better comparator. A comparison to regularized QDA might be more appropriate. Equally there have been several supervised implementation of PCA, including from Tibsharini's group (available in several R packages including superPC), Barshan et al., 2011 (<https://doi.org/10.1016/j.patcog.2010.12.015>) or between group analysis (available in the ade4 R package, <https://www.ncbi.nlm.nih.gov/pubmed/12490444>). Please compare your approach to one or more supervised PCA approaches

This methods performs a PCA of the difference in the covariance matrices. Does this approach work better than a PCA of results of a linear regression (e.g. limma). I am asking this as limma linear regression is widely performed by researchers to find features that separate two groups.

One of my principal concerns about this manuscript is the lack of reference to a rich resource of existing literature and methods in the field. Performing a decomposition of the differences between two matrices has been under-applied in genomics, however is widely used in other biological fields analysis. For example, distance based redundancy analysis, principal coordinates analysis and multiple dimensional scaling all are widely used (many implementations are described in classic textbook numerical ecological by Legendre & Legendre, 1998, or recently reviewed in Scientific Reports 7: 6481 (2017). [doi:10.1038/s41598-017-06693-z](https://doi.org/10.1038/s41598-017-06693-z)). Please compare your approach distance based matrix decomposition approaches.

The idea of separating a background and foreground population in data is useful and has been widely applied. Arguably one of the most popular background-foreground matrix decomposition approaches is projection pursuit. Please compare your approach to projection pursuit.

We thank the reviewer for the suggestion of comparing our technique to other related techniques. We have systematically compared our technique to all of the classes of techniques that the reviewer has suggested. In particular, we have compared **cPCA** to: **PCA**, **supervised PCA**, **LDA**, **QDA**, **limma (linear regression with PCA)**, **multidimensional scaling (MDS)**, **principal component pursuit (PCP)**, **factor analysis (FA)**, and **independent component analysis (ICA)**. We compared these techniques on two datasets in our paper: the synthetic dataset in Supp. Fig. 1 as well as the mice protein dataset. The code to reproduce these figures is in our GitHub repository. We have included these comparisons as supplementary figures in the manuscript [Supplementary Figures 3 and 4], and for convenience, we have reproduced these figures here.

Figure 2: 10 different dimensionality reduction techniques are applied to a synthetic dataset consisting of four subgroups. The resulting low-dimensional representation of the data is plotted here, with color corresponding to the subgroup that the data belongs to. (See GitHub repo for code)

Figure 3: 10 different dimensionality reduction techniques are applied to a mice protein dataset consisting of two subgroups. The resulting low-dimensional representation of the data is plotted here, with color corresponding to the subgroup that the data belongs to. (See GitHub repo for code)

In general, we find that existing techniques are unable to discover and resolve subgroups as effectively as cPCA. That cPCA discovers patterns different from those found by existing techniques is not surprising, as cPCA is able to utilize information from the background in a manner that is different from existing techniques, as we discussed in the response to (1).

In addition to these figures, we have expanded the “Related Works” section of our paper to address the reviewer’s concern about the lack of references in our original draft. It now reads (pg. 4-5):

cPCA is a tool for unsupervised learning, which efficiently reduces dimensionality to enable visualization and exploratory data analysis. This separates cPCA from a large class of supervised learning methods whose primary goal is to classify or discriminate between the various datasets. Examples include two-group comparison methods followed by a multiple testing procedure, like two-sample t-test, Wilcoxon signed-rank test, and Mann-Whitney U test, followed by Benjamini-Hochberg (BH) procedure. The family of classification and regression methods also fall into this category, e.g. linear discriminant analysis (LDA), quadratic discriminant analysis (QDA), supervised PCA, and QUADRO, where features important to the classifier are selected. All of these methods aim to identify features that differ in their means (or other statistic) between the target and background groups. While these differential features and statistics capture significant differences between the two datasets and can be used to predict whether a given data point comes from the target or the background set, they, unlike cPCA, do not try to discover patterns in the target data itself.

There is also a rich family of unsupervised methods for dimension reduction. For example, multi-dimensional scaling (MDS) finds a low-dimensional embedding that preserves the distance in the high-dimensional space; principal component pursuit finds a low-rank subspace that is robust to small entry-wise noise and gross sparse errors. But none are designed to utilize relevant information from a second dataset, as cPCA does. In the supplementary materials, we have compared cPCA to many of the previously-mentioned techniques on representative datasets (see Supp. Figures 3 and 4).

5. There have been other dimension reduction approaches that employ the difference in the covariance matrix describes in other fields. Please review if your methods is similar to (QUADRO), Fan et al, Ann Stat. 2015; 43(4): 1498–1534.) and Jin et al., 2005 (<https://dl.acm.org/citation.cfm?id=2118631>).

We thank the reviewer for pointing us to these techniques. After examining these techniques, we conclude that cPCA is quite different from both of these techniques.

QUADRO (Fan et al, 2015) is a dimensionality reduction technique that finds a non-linear low-dimensional space that maximally separates the two datasets (or the data from two classes). In this sense, its goal is similar to that of LDA/QDA, which is quite different from that of cPCA.

The covariance-based anomaly detection (Jin et al, 2005) is also quite different from our work. It computes covariance matrices for different classes of data, and to classify a new set of data, it compares its covariance matrix entry-wisely with the existing covariance matrices. In the setting where this technique is used, there are many classes of datasets, and the goal is to classify a new dataset into one of the existing classes. This is very different from the setting of cPCA.

6. How does homoscedasticity impact the performance of the approach? Is the approach robust to heteroscedasticity?

Generally, cPCA does not assume homoscedasticity in the data, in the sense that different features may have different levels of noise. Hence, we would argue that cPCA is robust to heteroscedasticity. cPCA does assume that the level of noise in each feature is consistent between the target and background dataset. This allows cPCA to distinguish between those features that have high variance in the target dataset because they are of interest to the analyst, versus those features that have high variance due to noise, as these features would also have high variance in the background dataset.

To demonstrate this concept, we chose a dataset that consisted of measurements from inertial measurement units strapped to a subject as he/she performed various activities. Because some sensors are noisier than other sensors, performing PCA on the unstandardized data identifies the *noisiest* features as those containing the most information, and so a PCA of the readings is not useful for separating the data points by activity. When cPCA is performed on the same dataset using, as a background dataset, readings from the same subject while he/she was lying still, cPCA learns to ignore the noisy sensors (since they remain noisy in the background dataset), allowing cPCA to effectively separate data points from the two activities.

The result is plotted in Figure 4. The code for this experiment can be found in the GitHub repository and the dataset is obtained from: <http://archive.ics.uci.edu/ml/datasets/mhealth+dataset>

Figure 4: We performed PCA and cPCA on a collection of IMU sensor recordings from a subject as he/she jogged (black) and squatted (red). Each data point represents sensor readings at one instant while the subject was performing the activities.

REVIEWER 2 COMMENTS:

1. This paper is well-written and very pleasant to read. I think it should be accepted after some minor revisions. My comments are following.

Thank you for your helpful feedback and for your support of our paper!

2. Is there any probabilistic interpretation to the cPCA approach? Perhaps analogous to the probabilistic PCA model Perhaps analogous to the probabilistic PCA model (see reference below).

Tipping, M. E., & Bishop, C. M. (1999). Probabilistic principal component analysis. *Journal of the Royal Statistical Society: Series B (Statistical Methodology)*, 61(3), 611-622.

Thank you for the suggestion. In the revision, we have added new analysis of cPCA in the context of a generative Gaussian model with white Gaussian noise; this setting is analogous to the probabilistic PCA model. We show that when α is small, cPCA seeks the maximum-variance direction that represents a trade-off between the subspace unique to the target data, and the subspace shared by the two datasets. When α exceeds a threshold, cPCA corresponds to performing PCA to the subspace unique to the target data. The details are now included in the supplementary materials, and we have reproduced them below.

4.3 A probabilistic interpretation

Suppose the target and the background data follow a Gaussian distribution. Then they can be written as linear combinations of standard Gaussian variables $Z_i, U_i, Z_{i'}, V_i \stackrel{i.i.d.}{\sim} \mathcal{N}(0, I)$, as well as noise vectors $\epsilon_i, \epsilon_{i'} \stackrel{i.i.d.}{\sim} \mathcal{N}(0, \sigma^2 I)$. The linear subspace can be determined as follows: Let $W_Y \in \mathbb{R}^{d \times p_Y}$ be the subspace unique to the background data, $W \in \mathbb{R}^{d \times p}$ be the rest of the subspace of the background data, and $W_X \in \mathbb{R}^{d \times p_X}$ be such that $W \cup W_X$ span the subspace of the target data. Then one can write

$$\begin{aligned} X_i &= WZ_i + W_X U_i + \epsilon_i \\ Y_{i'} &= WZ_{i'} + W_Y V_{i'} + \epsilon_{i'}, \end{aligned}$$

where we note that $\text{span}(W \cup W_X) \cap \text{span}(W_Y) = \emptyset$.

Let $W_{X,\perp}$ be the subspace of W_X that is perpendicular to the subspace $\text{span}(W)$ and let $W_{X,\parallel}$ be that parallel to $\text{span}(W)$. With some technical derivation, one can reach that

$$\mathbf{v}^* = \underset{\mathbf{v}}{\text{argmax}} \mathbf{v}^T \left(W_{X,\perp} W_{X,\perp}^T + W_{X,\parallel} W_{X,\parallel}^T + (1 - \alpha) W W^T \right) \mathbf{v}. \quad (1)$$

Now (1) is readily interpretable. When α is small, \mathbf{v}^* represents a trade-off between the space unique to the target data $\text{span}(W_{X,\perp})$ and the space shared between the two datasets $\text{span}(W)$. After α reach a threshold, \mathbf{v}^* becomes the first eigenvector of $W_{X,\perp} W_{X,\perp}^T$, i.e. the first principal component of the space unique to the target data. Specifically, in the special case when $\text{span}(W_X)$ is orthogonal to $\text{span}(W)$, this threshold is 1. In other words, when $\alpha \geq 1$, \mathbf{v}^* remains the first PC of the space unique to the target data, $\text{span}(W_{X,\perp})$.

3. I appreciate that the authors made the implementation available. Regarding the reproducibility of the real data analysis, I wonder if the data sets and data analysis code can be also be made available. Some of the analysis is difficult to reproduce exactly from the paper. For example, on page 7, ‘...before applying PCA or cPCA, all datasets are reduced to 500 genes, which are selected on the basis of highest dispersion within the target data’

We appreciate the reviewer’s concern for reproducibility. We have released the python notebooks with the code needed to reproduce our results, including the experiment to which the author refers. This code can be found in the GitHub repository for our paper: <https://github.com/abidlabs/contrastive/tree/master/experiments>. The notebooks correspond to Figures 2 and 3 in the main text, as well as Supplementary Figures 2-5.

4. The authors claimed that a “strong mathematical guarantees for the method.” But I do not find the theorems useful, since it failed to take into account any randomness in the data. In other words, I am wondering if there is any statistical guarantee like PCA or some other popular dimension reduction methods.

The cPCs correspond to the eigenvectors of the sample covariance matrix $(\hat{C}_X - \alpha \hat{C}_Y)$. Hence, the convergence property of the sample cPCs to the population cPCs is the same as that of the sample eigenvectors for a covariance matrix. We have added new finite sample statistical guarantee of cPCA by leveraging general results from the convergence of the eigenvectors of random matrices. Our statistical guarantee says that: Let d be the dimension and n be the sample

size. Under mild assumptions, the difference between sample cPC and population cPC scales with $O(\sqrt{d/n})$. We added the following paragraphs in the main article and the supplementary materials respectively:

Main article, theoretical guarantee section:

We also remark that regarding the randomness of the data, the convergence rate of the sample cPC to the population cPC is $O\left(\sqrt{\frac{d}{\min(n,m)}}\right)$ under mild assumptions, where d is the dimension and n, m are the size of the target and the background data. This rate is similar to the standard convergence rate of the sample eigenvector for a covariance matrix. See Supp. Sec. 4.2.

Supplementary material:

4.2 Convergence rate of the sample cPC

So far, the analysis concerns only the population cPC calculated based on the population covariance matrix $C = C_X - \alpha C_Y$. In practice, we only have finite number of data, say n data points from the target data and m data points from the background data. Let \hat{C}_X and \hat{C}_Y be the sample covariance matrices and let $\hat{C} = \hat{C}_X - \alpha \hat{C}_Y$. Then the sample cPC's are eigenvectors of \hat{C} . Here we characterize the convergence rate of the sample cPC to the population cPC.

Since the sample cPC corresponds to the eigenvector of the sample covariance matrix \hat{C} , the convergence property of the sample cPC is the same as that of the sample eigenvectors for a covariance matrix, which is well studied in previous literature [10, 11, 12]. In short, under mild assumptions, the sample cPC will converge to the population cPC when the data size is larger than the dimensionality. We state this formally in the following theorem.

Theorem 2. (*Convergence rate of the sample cPC*) Let $\hat{\mathbf{v}}^*$ be the first sample cPC and \mathbf{v}^* be the first population cPC². Assume that the entries of the target and the background data are sub-Gaussian with some fixed parameter, and the gap between the first and the second eigenvalue of C are bounded away from 0. Then with high probability,

$$1 - |(\hat{\mathbf{v}}^*)^T \mathbf{v}^*| = O\left(\sqrt{\frac{d}{\min(n, m)}}\right).$$

²Similar results can be shown for other cPC's by assuming an eigenvalue gap and using Wedin's Theorem.

REVIEWERS' COMMENTS:

Reviewer #1 (Remarks to the Author):

I enjoyed reading this well-written manuscript that presents a method that will be of general interest to the field. I thank for authors for carefully addressing my concerns and for comparing their method, cPCA to other approaches. The additional comparisons really helped place this importance of this method in context. Thank you for putting the code and data on github

1. One important attractive feature of this method it that the baseline or background data might be derived from a source other than the experiment data under study. On pg 14 in the section on choosing the background dataset, it states "cPCA does not require the target and background data to be similarly sized". This is a particularly attractive feature of this approach, frequently the number of features can vary study to study. May I asked if it would be appropriate to use an RNAseq dataset as a comparator/baseline dataset to analyze a proteomics dataset? Is there the minimum/max number of features/samples in the comparator/background/baseline dataset? Frequently in cancer research, the cell of origin is unknown, so the "normal" tissue comparator is unknown. For example the normal cell of origin of high grade serous ovarian cancer was only recently shown to be fallopian not ovarian epithelial. In other cases, only a few normal tissue samples are available, For example in the TCGA BRCA samples, there was RNAseq data on 1085 tumors, 114 normal samples (and 7 metastases), however there was no RPPA proteomics on the normal cases. This is an attractive feature of cPCA and should be highlighted in the abstract.

May I suggest the following edits that I think will assist maximize the biological and clinician impact of the approach

2. .Page 1. Abstract, I think this could be improved by changing a few words to make it clearer for biologists. A biologist would never consider the baseline and treated data to be "multiple datasets" The control data in integral to the study. Thus the used of the term "multiple datasets" may confuse biologists, they may consider multiple datasets to be data derived from different sources (eg RNAseq, v proteomics, or DNaseq). Biological experiments will always (or should always) include positive and negative control, and in the case of drug treatment, the negative control or baseline control are sham or untreated cells. May I suggest replacing "multiple datasets" with "multiple conditions to be compared". Frequently the term baseline, is used to refer to a control study and might be a good alternative term. Biological studies will normally consider the entire data generated in one experiment to be a dataset (all RNAseq from all samples, baseline, treatment, sham, treatment+inhibitor). I would suggest that the authors re-read the manuscript with this in mind, to see if the use of these words could be clarified.

3. Page 2 Introduction, paragraph 1. There are many methods, including numerous extensions of PCA that integrate >1 dataset and these have been WIDELY applied in genomics, however most of these seek to identify the covariant or correlated patterns among 2 or more datasets. Please refer to Meng et al., 2016 (<https://doi.org/10.1093/bib/bbv108>), Rohart et al., 2017 (<https://doi.org/10.1371/journal.pcbi.1005752>), Garali et al., 2017 (<https://doi.org/10.1093/bib/bbx060>), Stein-O'Brien, et al., (<https://www.biorxiv.org/content/early/2017/10/02/196915.1>) among others. Indeed it might be good to include a statement saying how cPCA differs to these integrative data approaches.

4. pg 6 results. second paragraph "we assume this is not known to the analyst" is inaccurate and does not reflect the nature of the experiments that generated these data. These mice were bred to

have trisomy, so I expect the genotype of these mice was known. Trisomy (3 chromosomes) is a mouse model of downs syndrome. The mice are subject to learning and the investigator would be using the trisomy mice as a negative control (they would be expected to learn slower). I would expect that these conditions would be known apriori. First please cite the original articles that generates these datasets rather than a subsequent article that described SOM. Within these data there are eight groups of mice based on genotype (WT, trisomy), stimulation/shock to learn (Context-Shock, Shock-Context) and treatment (sham (saline), memantine).

5. In suppl Fig 2A, there is little separation of data on PC1, $\alpha=0$ and all values are negative. Typically, when the features on PC1 have all negative or all positive weights, this is an artifact of data indicating that data were not normalized. How were these data normalized?. In suppl Fig 4, it appears that PC1 (PC1, sPCA, FA) etc all have weights on one side of the origin. I downloaded these data, and the effect is removed by applying a normalization (which you argue correctly is not always appropriate) or by applying a dual scaling matrix decomposition methods such as correspondence analysis.

6. In your code, when you run ICA, you select two components. With ICA, the components are not ranked (this is a disadvantage of the approach), so you would need to extract >2 components and test for clusters (or assoc with known covariate)

7. The results from the simulated data do not appear to be consistent with the real data. Do the simulated data have the correlation structure expected in 'omics data. Genomics data typically have strong correlation structure among features (compare a matrix of the Euclidean distance v 1-Pearson Correlation). The magnitude of change in feature levels may be of little biological importance. Some genes/proteins that are constitutively active genes, these will be highly expressed, and are on in most cells. Whereas some genes such as immune genes or transcription factors may have strong biological response to small changes in magnitude of expression. Therefore, Pearson correlation is often used in preference to Euclidean distance. However generating simulated data that really reflect the complexity in genomics data is challenging, there I would just mention the limitations of the simulated data used.

Reviewer #2 (Remarks to the Author):

I have no further comment.

We thank the reviewers for their thoughtful feedback. We provide point-by-point response to all of the reviewers' questions below.

REVIEWER 1 COMMENTS:

1. One important attractive feature of this method is that the baseline or background data might be derived from a source other than the experiment data under study. On pg 14 in the section on choosing the background dataset, it states "cPCA does not require the target and background data to be similarly sized". This is a particularly attractive feature of this approach, frequently the number of features can vary study to study. May I ask if it would be appropriate to use an RNAseq dataset as a comparator/baseline dataset to analyze a proteomics dataset? Is there the minimum/max number of features/samples in the comparator/background/baseline dataset? Frequently in cancer research, the cell of origin is unknown, so the "normal" tissue comparator is unknown. For example the normal cell of origin of high grade serous ovarian cancer was only recently shown to be fallopian not ovarian epithelial. In other cases, only a few normal tissue samples are available, For example in the TCGA BRCA samples, there was RNAseq data on 1085 tumors, 114 normal samples (and 7 metastases), however there was no RPPA proteomics on the normal cases. This is an attractive feature of cPCA and should be highlighted in the abstract.

Contrastive PCA allows the target data and the background data to be derived from different sources and have different sample sizes. However, we have currently developed it and evaluated it for cases where the features are the same in the background and the target datasets. Nonetheless, it is a future research direction to consider datasets that do not share exactly the same set of features.

A general guidance regarding the feature size and the sample size is that they should be at least comparable. As is the case for PCA, performance generally improves if the sample size is at least a few times larger than the features size. In RNAseq experiments, this can be achieved by filtering out unrelated genes prior to the cPCA analysis based on some simple criterion, e.g. the coefficient of variation.

2. Page 1. Abstract, I think this could be improved by changing a few words to make it clearer for biologists. A biologist would never consider the baseline and treated data to be "multiple datasets" The control data is integral to the study. Thus the use of the term "multiple datasets" may confuse biologists, they may consider multiple datasets to be data derived from different sources (eg RNAseq, v proteomics, or DNaseq). Biological experiments will always (or should always) include positive and negative control, and in the case of drug treatment, the negative control or baseline control are sham or untreated cells. May I suggest replacing "multiple datasets" with "multiple conditions to be compared". Frequently the term baseline, is used to refer to a control study and might be a good alternative term. Biological studies will normally consider the entire data generated in one experiment to be a dataset (all RNAseq from all samples, baseline,

treatment, sham, treatment+inhibitor). I would suggest that the authors re-read the manuscript with this in mind, to see if the use of these words could be clarified.

Thanks for the suggestion. We have revised the abstract and the introduction is as follows:

In the abstract:

However, in many settings we have datasets collected in different conditions,

In the introduction,

When the analyst has multiple datasets (or multiple conditions in one dataset to compare), then the current state-of-practice is to perform PCA (or t-SNE, MDS, etc.) on each dataset separately, and then manually compare the various projections to explore if there are interesting similarities and differences across datasets

3. Page 2 Introduction, paragraph 1. There are many methods, including numerous extensions of PCA that integrate >1 dataset and these have been WIDELY applied in genomics, however most of these seek to identify the covariant or correlated patterns among 2 or more datasets. Please refer to Meng et al., 2016 (<https://doi.org/10.1093/bib/bbv108>), Rohart et al., 2017 (<https://doi.org/10.1371/journal.pcbi.1005752>), Garali et al., 2017 (<https://doi.org/10.1093/bib/bbx060>), Stein-O'Brien, et al., (<https://www.biorxiv.org/content/early/2017/10/02/196915.1>) among others. Indeed it might be good to include a statement saying how cPCA differs to these integrative data approaches.

Thank you for the suggestion. We have cited the papers and compared them with cPCA in the introduction.

4. 4. pg 6 results. second paragraph "we assume this is not known to the analyst" is inaccurate and does not reflect the nature of the experiments that generated these data. These mice were bred to have trisomy, so I expect the genotype of these mice was known. Trisomy (3 chromosomes) is a mouse model of downs syndrome. The mice are subject to learning and the investigator would be using the trisomy mice as a negative control (they would be expected to learn slower). I would expect that these conditions would be known apriori. First please cite the original articles that generates these datasets rather than a subsequent article that described SOM. Within these data there are eight groups of mice based on genotype (WT, trisomy), stimulation/shock to learn (Context-Shock, Shock-Context) and treatment (sham (saline), memantine).

We agree with the reviewer that information about (Down Syndrome) DS is known *a priori* for this dataset. The reason we formulate the problem in this way is to create an unsupervised learning task. which we can use evaluate cPCA relative to PCA and other

unsupervised learning algorithms. The DS labels are used to evaluate the algorithms afterwards to see which dimensionality reduction techniques best separated the DS and non-DS mice. We have revised the manuscript to make this clearer:

To create an unsupervised learning task where we have ground truth information to evaluate the methods, we assume this DS information is not known to the analyst and only use it for algorithm evaluation. We would like to see if we detect any significant differences within the shocked mice population in an unsupervised manner (the presence or absence of Down Syndrome being a key example).

5. In suppl Fig 2A, there is little separation of data on PC1, $\alpha=0$ and all values are negative. Typically, when the features on PC1 have all negative or all positive weights, this is an artifact of data indicating that data were not normalized. How were these data normalized?. In suppl Fig 4, it appears that PC1 (PC1, sPCA, FA) etc all have weights on one side of the origin. I downloaded these data, and the effect is removed by applying a normalization (which you argue correctly is not always appropriate) or by applying a dual scaling matrix decomposition methods such as correspondence analysis.

We would like to point out in the first panel of Supp Fig 2a, the data are actually centered around 0. The reason that the axis values are skewed negative is that there are a few outliers with extremely negative values.

The same phenomenon happens in Supp Fig 4. For the panels mentioned by the reviews, the majority of the data are well-centered around 0. The negative skew on the x-axis values is due to the few outliers.

This does not happen in Supp. Fig 3 because the simulation data does not have outliers.

6. In your code, when you run ICA, you select two components. With ICA, the components are not ranked (this is a disadvantage of the approach), so you would need to extract >2 components and test for clusters (or assoc with known covariate)

Thanks for pointing out. We rerun ICA with 10 components and plotted the results of each dimension as below. As we can see, none of the components discover the sub-cluster structure. So adding more components does not seem to improve the performance.

Moreover, given the components, it is not clear how to choose the components in an unsupervised fashion. First, since it is unsupervised, the labels are not available. For methods that test for cluster, the elbow method is most widely-used. However, it involves the interaction with the analyst and cannot be used here. An alternative is the gap statistics [Tibshirani et al, 2001], but this is not widely agreed upon. Due to these considerations, we believe that it would be better to just keep the original plot with 2 components.

- The results from the simulated data do not appear to be consistent with the real data. Do the simulated data have the correlation structure expected in 'omics data. Genomics data typically have strong correlation structure among features (compare a matrix of the Euclidean distance v 1- Pearson Correlation). The magnitude of change in feature levels may be of little biological importance. Some genes/proteins that are constitutively active genes, these will be highly expressed, and are on in most cells. Whereas some genes such as immune genes or transcription factors may have strong biological response to small changes in magnitude of expression. Therefore, Pearson correlation is often used in preference to Euclidean distance. However generating simulated data that really reflect the complexity in genomics data is challenging, there I would just mention the limitations of the simulated data used.

Thanks for the suggestion. We have now included the limitations of the simulated data in the caption of Supp. Fig. 1 (where the data first appeared) as follows:

After this, a random rotation is applied to the both datasets to make it non-trivial to find the directions that separate the subgroups in the target dataset. We remark

that the purpose of the synthetic dataset is to demonstrate the behavior of cPCA; it is not meant to capture the complexity found in the structure of biological (e.g. genomic) datasets.

REVIEWER 2 COMMENTS:

1. I have no further comment.

Thank you for your helpful feedback and for your support of our paper!